# Deep learning-based approach to the characterization and quantification of histopathology in mouse models of colitis

Soma Kobayashi[1], Jason Shieh[2], Ainara Ruiz de Sabando[3], Julie Kim[2], Yang Liu[2], Sui Y. Zee[4], Prateek Prasanna[1], Agnieszka B. Bialkowska[2], Joel H. Saltz[1,4], Vincent W. Yang[1,2,5]*

1 Department of Biomedical Informatics, Stony Brook University, Stony Brook, NY, United States of America, 2 Department of Medicine, Renaissance School of Medicine at Stony Brook University, Stony Brook, NY, United States of America, 3 Department of Medical Genetics, Complejo Hospitalario de Navarra, Pamplona, Navarra, Spain, 4 Department of Pathology, Renaissance School of Medicine at Stony Brook University, Stony Brook, NY, United States of America, 5 Department of Physiology and Biophysics, Stony Brook University, Stony Brook, NY, United States of America

* Vincent.Yang@stonybrookmedicine.edu

**Data Availability Statement:** All relevant data are within the manuscript and its Supporting Information files.

## Abstract

Inflammatory bowel disease (IBD) is a chronic immune-mediated disease of the gastrointestinal tract. While therapies exist, response can be limited within the patient population. Researchers have thus studied mouse models of colitis to further understand pathogenesis and identify new treatment targets. Flow cytometry and RNA-sequencing can phenotype immune populations with single-cell resolution but provide no spatial context. Spatial context may be particularly important in colitis mouse models, due to the simultaneous presence of colonic regions that are involved or uninvolved with disease. These regions can be identified on hematoxylin and eosin (H&E)-stained colonic tissue slides based on the presence of abnormal or normal histology. However, detection of such regions requires expert interpretation by pathologists. This can be a tedious process that may be difficult to perform consistently across experiments. To this end, we trained a deep learning model to detect 'Involved' and 'Uninvolved' regions from H&E-stained colonic tissue slides. Our model was trained on specimens from controls and three mouse models of colitis–the dextran sodium sulfate (DSS) chemical induction model, the recently established intestinal epithelium-specific, inducible $Klf5^{\Delta IND}$ ($Villin\text{-}CreER^{T2};Klf5^{fl/fl}$) genetic model, and one that combines both induction methods. Image patches predicted to be 'Involved' and 'Uninvolved' were extracted across mice to cluster and identify histological classes. We quantified the proportion of 'Uninvolved' patches and 'Involved' patch classes in murine swiss-rolled colons. Furthermore, we trained linear determinant analysis classifiers on these patch proportions to predict mouse model and clinical score bins in a prospectively treated cohort of mice. Such a pipeline has the potential to reveal histological links and improve synergy between various colitis mouse model studies to identify new therapeutic targets and pathophysiological mechanisms.

**Funding:** NIH grants: DK052230 to V.W.Y. and CA205109/CA225021 to J.H.S. The funders had no role in study design, data collection and analysis, decision to publish, or preparation of the manuscript.

**Competing interests:** The authors have declared that no competing interests exist.

## Introduction

Inflammatory bowel disease (IBD) is a state of chronic intestinal inflammation that is comprised of two major subtypes, Crohn's disease (CD) and ulcerative colitis (UC). IBD afflicts approximately 1.6 million Americans, and as many as 70,000 new cases are diagnosed each year [1]. In addition to intestinal symptoms such as abdominal pain and diarrhea, patients can be severely affected by extraintestinal manifestations such as arthritis, ankylosing spondylitis, erythema nodosum, pyoderma gangrenosum, iritis, uveitis, and primary sclerosing cholangitis [2]. Furthermore, CD has been linked to increased risk for cancer, pulmonary, gastrointestinal, genital, and urinary tract diseases [1], and UC is associated with an increased risk for colorectal cancer [3]. Classically, CD and UC have different patterns of intestinal involvement. CD pathology is discontinuous and can affect any area along the gastrointestinal tract, while UC has a continuous distribution that is limited to the colon [4]. Both diseases can have uninvolved intestinal regions. In CD, affected areas are termed 'skip lesions' as they are interspersed with uninvolved regions. In UC, disease often does not involve the whole large intestine.

Mouse models of colitis have been heavily studied to understand disease pathogenesis and identify new treatment targets. Colons from these mice also have involved and uninvolved regions. One of the most used, in part due to the relatively simple method of induction and replicability, is the dextran sodium sulfate (DSS) chemical induction model. DSS is provided to mice in drinking water for seven days to cause acute colitis [5]. DSS-treated mice have a proximal-sparing injury pattern [6, 7], and Kolachala et al. reported different cytokine profiles between the proximal and distal colon [6]. Studies have characterized the immune response in these mice with flow cytometry and single-cell RNA [8, 9]. However, these protocols require the processing of whole colons at once. Diseased and healthy colonic regions are thus mixed before dissociation into single cells. Such an approach may not be sufficient to fully capture intracolonic heterogeneity. Our group has also observed the simultaneous presence of colonic regions involved or uninvolved with disease in the recently established the $Klf5^{\Delta IND}$ colitis mouse model [10, 11]. Therefore, we were motivated to develop an automated method to histologically identify these areas across these mouse models.

Convolutional neural networks (CNNs) have shown much promise in biomedical image classification tasks. CNNs learn to associate visual patterns with image labels [12–14]. Histologic findings are labels that pathologists have attributed to common cellular, morphological, and tissue patterns associated with disease. Since histopathological analysis is grounded upon detection of such labels, CNNs are well-suited for this domain. In practice, histological slides are digitized into gigapixel resolution whole slide images (WSIs). Due to the file sizes of WSIs, they are broken up into smaller image patches to which CNNs are applied. Patch-level, CNN outputs can be aggregated to make WSI-level predictions. In this study, we incorporated CNNs in a pipeline that quantifies patch-level histological findings to characterize murine colon WSIs.

Novel protein and genomic targets are frequently identified in preclinical animal models for assessment in clinical studies. In contrast, use of histology is often limited to confirming presence or absence of these targets. Characterization and quantification of histological patterns is often not a focus in preclinical mouse models. However, histological patterns are clinically significant and a manifestation of cellular and sub-cellular mechanisms mediated by protein and genomic targets. This is evidenced by the clinical impact of polyp classes, as the extent of the "villous" subtype is a major independent risk factor for high-grade dysplasia [15]. There is therefore likely value in mapping the presence of histological patterns across colitis mouse models to molecular studies. Adaptation of preclinically-defined histological phenotypes to routinely collected clinical specimens has the potential for more spatially-granular, non-invasive characterizations.

An automated pipeline to detect intracolonic heterogeneity could significantly decrease time needed to characterize histology of colitis mouse models. This could circumvent the need for a pathologist to undergo a tedious scoring process. The histological readouts could also aid in the evaluation of treatment efficacies. In addition, reliable identification of diseased and healthy colonic areas is the first step towards further phenotyping cellular populations by spatial context. Cell populations can be mapped immunohistochemically to those regions to provide a spatial context-aware characterization of immune responses. Lastly, our computational approach can extract predicted image regions across a mouse cohort. 'Involved' and 'Uninvolved' region extraction allows for clustering approaches to identify histological classes. We quantified the proportion of 'Uninvolved' patches and 'Involved' patch classes to train separate machine learning classifiers for two tasks. In our prospectively treated cohort of mice, these two classifiers predicted mouse model and clinical score bins with overall F1 scores of 95.75% and 86.39%, respectively.

## Results

### Archived mouse cohort background

To train our models, we utilized archived, formalin-fixed paraffin-embedded (FFPE) swiss-rolled colons [16]. Whole colons were collected from three colitis mouse models and appropriate controls. The colitis model treatment schedules are detailed in S1A Fig. The first mouse model is the recently established $Klf5^{\Delta IND}$ ($Villin-CreER^{T2};Klf5^{fl/fl}$) genetic model. Inducible intestinal epithelium-specific knockout of $Klf5$ upon five days of intraperitoneal (IP) tamoxifen (TAM) injections disrupts epithelial barrier function and causes colitis (5T-$Klf5^{\Delta IND}$) [10, 11]. We utilized female mice due to the higher rate of $Klf5$ knockout relative to males [11]. As TAM is dissolved in corn oil (CO), we also performed five days of IP CO injections in $Klf5^{\Delta IND}$ mice as a control (5C-$Klf5^{\Delta IND}$). Additionally, we collected various combinations of biological controls with five days of IP CO and five days of IP TAM injections across $Klf5^{\Delta IND}$, $Klf5^{\Delta IND/+}$ ($Villin-CreER^{T2};Klf5^{fl/+}$), and $Klf5^{WT}$ mice ($Villin-CreER^{T2};Klf5^{+/+}$) (S1 Table).

The second mouse model is the chemical dextran sodium sulfate (DSS) induction model. To control for the injections in our 5T-$Klf5^{\Delta IND}$ model, $Klf5^{\Delta IND/+}$ mice received five days of IP CO injections. This was followed by seven days of 2.5% DSS in drinking water. The histology mirrors that in non-injected $Klf5^{WT}$ mice treated with DSS (S1B Fig). For the third model, we performed a combined induction. As homozygous 5T-$Klf5^{\Delta IND}$ mice exhibit increased mortality in an 18-day period after induction [11], we injected heterozygous $Klf5^{\Delta IND/+}$ mice with TAM for five days IP, then followed with seven days of 2.5% DSS in drinking water (5T-$Klf5^{\Delta IND/+}$ + DSS). Although we have yet to fully characterize the combined induction model, we wanted to examine the histology in these mice. Specifically, we evaluated whether our computational pipeline could distinguish the single-type induction models even with these mice in the cohort. We gathered all relevant archived samples that were collected by past lab members. In total, we have 48 mice in the archived mouse cohort. Sample numbers are available in S1 Table.

Colons extracted from these colitis models contain areas that are involved or uninvolved with disease. Manually selected regions from representative whole slide images (WSIs) are shown in Fig 1A. Swiss rolls allow for the visualization of a whole mouse colon on a single glass side. In these preparations, the center is the proximal end. The colon tissue can be traced distally towards the outer portion of the swiss roll. Although not utilized to train our model, we provide WSIs of additional biologic controls (5C-$Klf5^{\Delta IND/+}$ and 5T-$Klf5^{\Delta IND/+}$) for the combined induction model (S1C Fig). Clinical scoring metrics combining weight loss, stool consistency, and fecal blood [17] trend higher for 5T-$Klf5^{\Delta IND/+}$ + DSS mice relative to control

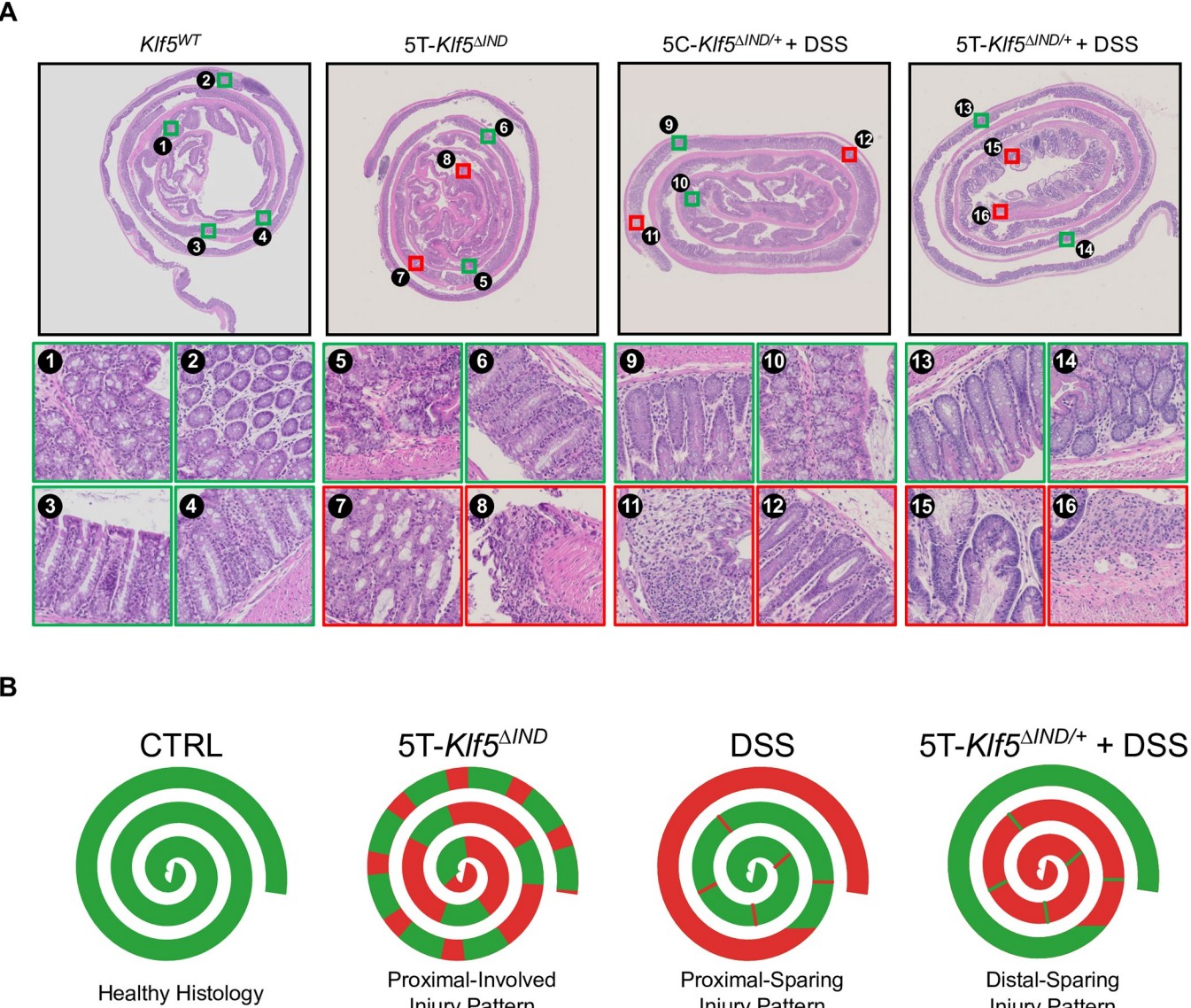

**Fig 1. Colitis mouse models exhibit regional heterogeneity.** A) Representative whole slide images (WSIs) of swiss–rolled colons from *Klf5*$^{WT}$ (*Villin–CreER*$^{T2}$;*Klf5*$^{+/+}$) control and colitis mouse models. Manually selected regions that are 'Involved' (red) and 'Uninvolved' (green) are shown for colitis mouse model samples. 8, 11, and 16 show areas with crypt dropout. 7 and 12 are examples of crypt dilation. 15 is an example of distorted glands. B) Observed patterns of injury for colitis mouse models.

and 5C-*Klf5*$^{\Delta IND/+}$ + DSS mice (S1D Fig). 5T-*Klf5*$^{\Delta IND/+}$ + DSS mice trend lower on histological scoring relative to 5C- *Klf5*$^{\Delta IND/+}$ + DSS mice (S1E Fig), mainly due to decreased extent of ulceration along the colon. This mismatch in clinical and histological scoring trends helped motivate the development of our automated pipeline.

While the regional heterogeneity in our colitis models is visually apparent (Fig 1A), identification of these regions requires a pathologist's inspection. This can be difficult at times to perform objectively and especially over many samples. Pathologist inter- and intra-observer variability has been reported in various contexts, such as lymph node counting [18], atypical ductal hyperplasia diagnosis [19], and follicular thyroid carcinoma diagnosis [20]. An automated pipeline to identify murine colonic regions involved and uninvolved with disease would

provide benefits in objective reproducibility and speed. Furthermore, reliable identification of these areas opens the door for more spatially motivated, histological characterizations. A goal of this pipeline is to capture the reported proximal-involving and proximal-sparing injury patterns in the 5T-*Klf5$^{\Delta IND}$* [11] and DSS models [6, 7], respectively (Fig 1B). To this end, we trained an 'Involved' versus 'Uninvolved' classification model that generates predictions over mucosal and submucosal regions of swiss-rolled murine colonic tissues.

## 'Involved' versus 'Uninvolved' classifier training

For our classifier, we used the ResNet-34 (RN-34) architecture [21]. ResNet is considered a landmark architecture for the introduction of skip connections. Deep networks introduce a high number of non-linear mathematical operations through the addition of many layers. This can complicate the gradient descent calculations that allow the neural network to "learn". Gradient explosion and vanishing occur when calculated gradients are too large or small and will impede learning [22]. Skip connections allow gradient descent to skip portions of the network where this occurs to continue training. This has made possible a deep, 34-layer ResNet that can extract even higher dimensional features from images, a powerful quality for H&E-patch classification. We trained RN-34 models that were pretrained on ImageNet, a large dataset with natural images comprising 1000 classes [23].

Our classifier was trained in a two-phase approach (S2A and S2B Fig). All WSIs from our archived mouse cohort were downsampled by a factor of 8 and tiled into equally sized 224x224 pixel patches. The initial phase classifier was trained with ground truth patch labels corresponding to mouse colitis status (S2A Fig). As such, all patches from colitis mice were labeled 'Colitis' and those from control mice were labeled 'Control'. This labeling process does not account for the intracolonic heterogeneity observed in our colitis models (Fig 1A). Therefore, we aimed to generate patch-level ground truth labels to improve model performance. We used our initial phase model as a feature extractor to collect the high dimensional patch representations learned during the initial phase of training (S2B Fig). K-means clustering on these representations across all archived mouse cohort samples identified five patch classes (S2C and S2D Fig). Three of these ('Crypts', 'Lightly Packed', and 'Rosettes') were visually determined and validated with a pathologist to be uninvolved, while the other two ('Mixed Pathology', 'Distorted Glands') were involved with abnormal histology.

We then quantified these patch classes across our mice. For each mouse, we calculated the proportion of each of the five patch classes out of the total number of patches from the swiss roll. Two of the three qualitatively uninvolved classes were significantly enriched in our control mice relative to colitis mice, while both of our involved classes were significantly enriched in colitis mice relative to controls (S2D Fig). The lack of significance for the 'Rosettes' class is attributed to the higher proportion of this patch type in DSS-treated mice relative to the other colitis models (S2E Fig). This is likely due to the proximal-sparing injury pattern in DSS-treated mice [6, 7], as rosette structures are more prevalent in the proximal colon. The increased proximal prevalence is from sectioning mucosal folds, which are found in the proximal murine colon [24, 25]. In the second phase of training (S2B Fig), all patches from control mice were labelled 'Uninvolved' as no colitis induction occurred. All patches from colitis mice were labelled 'Uninvolved' or 'Involved' according to k-means patch class.

In both phases of training, 70% of total patches were used to train the model, 10% as a mid-training performance validation set, and 20% as a held-out test set. Our final model exhibited an overall F1 score of 90.1% on the held-out test set (Fig 2A). This was an improvement from the initial phase model (overall F1 score of 77.6%) that was trained with colitis status ground truth labels (S2F Fig). Notably, the initial phase model exhibited a relatively lower F1 score of

68.87% when attempting to predict patches from colitis mice. This performance is likely explained by the uninvolved patches from colitis mice labeled as 'Colitis' and motivated our second phase of model training. Additionally, when applied to 171 patches with 'Involved' and 'Uninvolved' ground truth labels provided by a pathologist, our final classifier exhibited an overall F1 score of 81.41% and again showed improvement from the initial phase model (S2G Fig).

## Small patch classifier training for preprocessing

We initially used simple thresholding of pixel intensities across each patch's red, green, and blue color channels to filter out patches with too much background and muscle. However, this approach was not robust. We trained an additional RN-34 classifier pretrained on ImageNet [23] on smaller 32x32 pixel patches to classify between 'Background', 'Muscle', 'Tissue', and 'Submucosa' classes (S3 Fig). From 8 mice (2 each of control and of the three mouse models), we extracted 100 patches for each of the 'Background', 'Muscle', 'Tissue', and 'Submucosa' classes (S3A Fig). Our model was trained with these ground truth labels and applied to a test set of patches from 4 separate mice (1 each of control and the 3 mouse models, 100 patches for each class). This model achieved an overall F1 score of 93.5% (S3B Fig). We also generated qualitative overlays of our small patch classifier's prediction on test set mice (S3C Fig). Additionally, we implement this small patch classifier during our patch filtering process. All patches with >65% of area corresponding to regions predicted to be 'Background' or 'Muscle' are filtered out during the patch extraction process (S4 Fig). We identified 65% as the ideal cutoff to ensure proper coverage of mucosal areas across swiss-rolls, while filtering out patches with too much 'Muscle' or 'Background' regions.

## 'Involved' versus 'Uninvolved' overlay generation

We generated overlays as visual outputs for our classifications. We sought to first reduce 'Involved' versus 'Uninvolved' classification dependence on what patch happened to be extracted from a region. We extracted overlapping patches by taking initial patch locations and iterating 20 pixels 10 times in each direction (up/down/left/right). This led to an increase of ~450 to ~200,000 patches per mouse after repeating patch filtering. Our classifier was applied to each overlapping patch, and 'Involved' and 'Uninvolved' prediction confidences were averaged at every pixel in the WSI. The prediction associated with each pixel thus considers more spatial context than just the initial patch it resided in. Areas with at least 50% confidence of 'Involved' and 'Uninvolved' predictions are overlayed onto the input H&E WSIs (Fig 2B). All regions predicted by our Small Patch Classifier to be 'Background' and 'Muscle' were eliminated from final overlays.

## Prospective mouse cohort

To test the robustness of our 'Involved' versus 'Uninvolved' Classifier, we treated and collected H&E-stained, FFPE swiss-rolled colons from 24 additional mice. Specifically, this cohort allowed us to assess whether our approaches would work regardless of who treated mice, collected swiss rolls, and performed H&E staining. This consisted of eight controls (no injections, put on normal drinking water), eight $Klf5^{\Delta IND/+}$ DSS-treated mice with no injections, five 5T-$Klf5^{\Delta IND}$, mice and three 5T-$Klf5^{\Delta IND/+}$ + DSS mice (S1 Table). We elected to utilize non-injected DSS-treated mice to confirm that our model properly captures the histology even without five days of corn oil injections preceding the DSS. We applied our approach to these prospective samples and generated 'Involved' versus 'Uninvolved' overlays (Fig 2C).

**A**

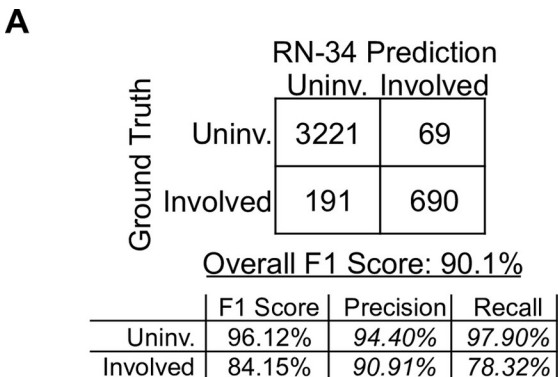

RN-34 Prediction

| Ground Truth | | Uninv. | Involved |
|---|---|---|---|
| | Uninv. | 3221 | 69 |
| | Involved | 191 | 690 |

Overall F1 Score: 90.1%

| | F1 Score | Precision | Recall |
|---|---|---|---|
| Uninv. | 96.12% | 94.40% | 97.90% |
| Involved | 84.15% | 90.91% | 78.32% |

**B**

Archived Cohort

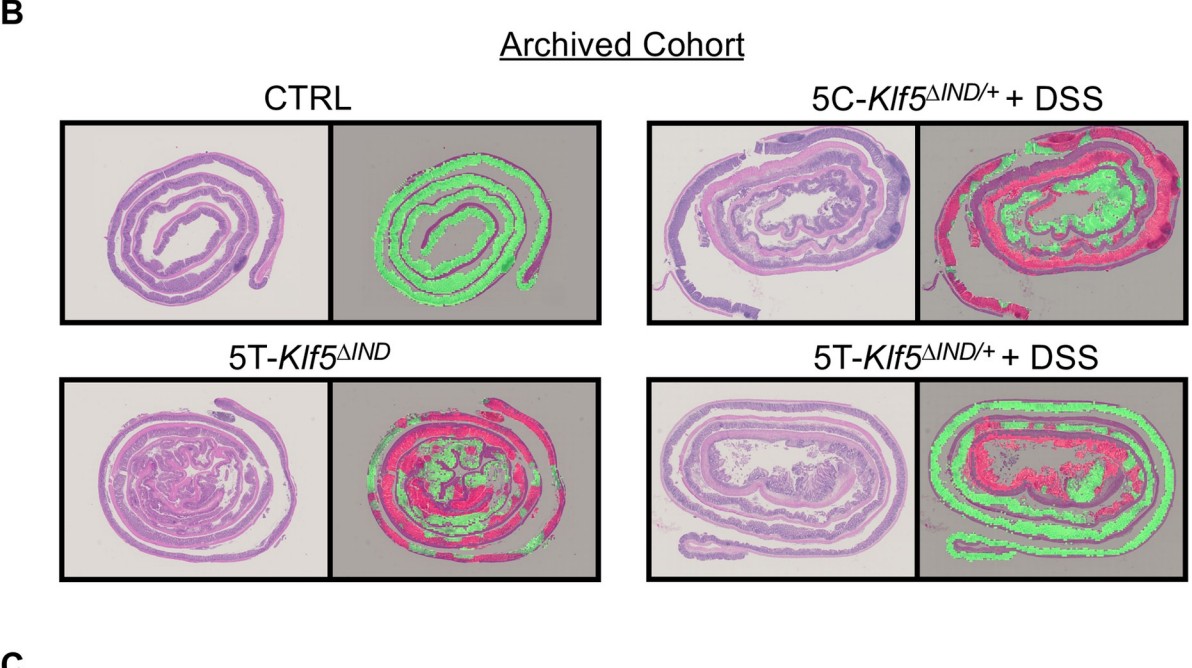

CTRL          5C-$Klf5^{\Delta IND/+}$ + DSS

5T-$Klf5^{\Delta IND}$          5T-$Klf5^{\Delta IND/+}$ + DSS

**C**

Prospective Cohort

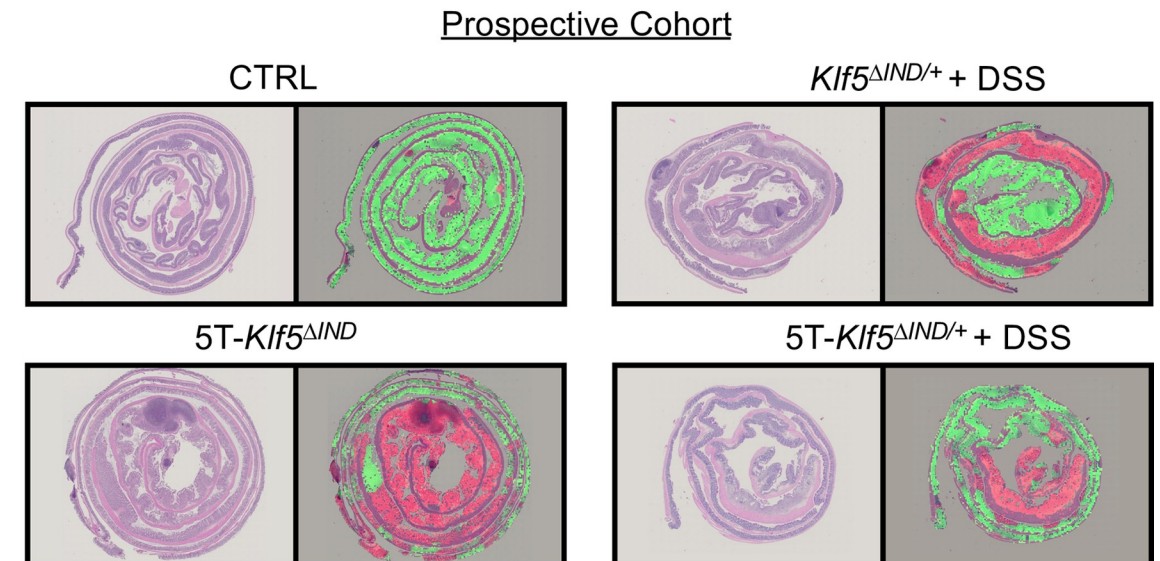

CTRL          $Klf5^{\Delta IND/+}$ + DSS

5T-$Klf5^{\Delta IND}$          5T-$Klf5^{\Delta IND/+}$ + DSS

**Fig 2. Classifier detects 'Involved' versus 'Uninvolved' regions in swiss–rolled colons.** A) Confusion matrix showing final model predictions on patches from test set mice. From the 48 mice in our archived cohort (S1 Table), 34 mice (14281 patches, ~70% of total) were used for model training, 5 mice were used for mid–training validation (2192 patches, ~10% of total), and 9 mice (4171 patches, ~19%) were used as a final independent test set. B) Original input H&E–stained swiss rolls are shown with corresponding 'Involved' (red) versus 'Uninvolved' (green) classifier overlays for archived mouse cohort test set mice. C) Original input H&E–stained swiss rolls and corresponding overlays for prospective mouse cohort samples. For both B) and C), red and green overlay colors correspond to regions with at least 50% prediction confidence.

## Patch class discovery

To build upon the binary 'Involved' and 'Uninvolved' predictions, we defined more specific image patch classes. Analogous to our second phase of model training (S2B Fig), we utilized our final 'Involved' versus 'Uninvolved' Classifier to extract high dimensional representations for patches in our archived mouse cohort. Specifically, two datasets (one with all 'Involved' patch representations and one with all 'Uninvolved' patch representations) were collected. Given the high number of patches, principal component analysis (PCA)-based dimensionality reduction was performed on each dataset before k-means clustering. Clustering on 'Uninvolved' patch representations identified three classes: 'Crypts' (test-tube-like perspective of crypt structures), 'Lightly Packed' (more space between crypts often accompanied by immune cell nuclei), and 'Rosettes' (Ring-like cross-section perspective of crypts) (Fig 3A and 3B). Of note, the discovered 'Uninvolved' patch classes matched those identified during our second phase of classifier training (S2C Fig).

While the 'Crypts' class was significantly enriched in control mice relative to all the colitis mouse models, this was not true for the other two classes. Specifically, the 'Rosettes' class was significantly enriched in control mice relative to 5T-$Klf5^{\Delta IND}$ and 5T-$Klf5^{\Delta IND/+}$ + DSS mice, but not to 5C-$Klf5^{\Delta IND/+}$ + DSS mice (Fig 3B). In addition, the 'Rosettes' class was significantly enriched in the 5C-$Klf5^{\Delta IND/+}$ + DSS mice relative to the other two colitis models. This is again likely due to the proximal-sparing injury pattern in DSS-treated mice [6, 7]. This injury pattern is also depicted in our 'Involved' versus 'Uninvolved' overlays (Fig 2B and 2C). Our DSS-treated mice show more 'Uninvolved' predictions in the rosettes-enriched proximal colon (center of swiss rolls) where disease is less common. In the 5T-$Klf5^{\Delta IND/+}$ + DSS mice, however, DSS-treatment no longer causes a proximal-sparing injury pattern (Figs 1B, 2B and 2C). While the mechanism of DSS-induced colitis has not been completely elucidated, the effects are believed to be dependent on tissue penetration of the chemical leading to disruption of the intestinal epithelial monolayer and barrier integrity [26]. DSS has a variable molecular weight from 5 to 1400 kDa, and administration of forms 500 kDa and higher do not induce colitis [27]. The shift in the proximal-sparing pattern of 5C-$Klf5^{\Delta IND/+}$ + DSS mice to the distal-sparing one in 5T-$Klf5^{\Delta IND/+}$ + DSS mice may be a function of increased proximal tissue DSS penetrance following TAM-induction.

The 'Lightly Packed' class was significantly enriched in control mice relative to 5C-$Klf5^{\Delta IND/+}$ + DSS mice but not the other two colitis mouse models. This likely relates to the increased prevalence of abnormal proximal histology in the 5T-$Klf5^{\Delta IND}$ and 5T-$Klf5^{\Delta IND/+}$ + DSS models relative to 5C-$Klf5^{\Delta IND/+}$ + DSS mice [11] (Fig 2B and 2C). However, the 'Crypts', which are found more distally, significantly decreased in all colitis mouse models relative to controls. Thus, an additional explanation is that the 'Lightly Packed' regions sit on the decision border. They may capture an accumulation of immune cells that is protective or too low a grade of abnormal histology to garner an 'Involved' prediction. While our 5T-$Klf5^{\Delta IND/+}$ + DSS model trended higher in clinical score than our 5C-$Klf5^{\Delta IND/+}$ + DSS mice (S1D Fig), they trended lower in histological scoring (S1E Fig). Scoring schemes that rely upon counting of pre-defined pathological findings without accounting for lower grade abnormalities may

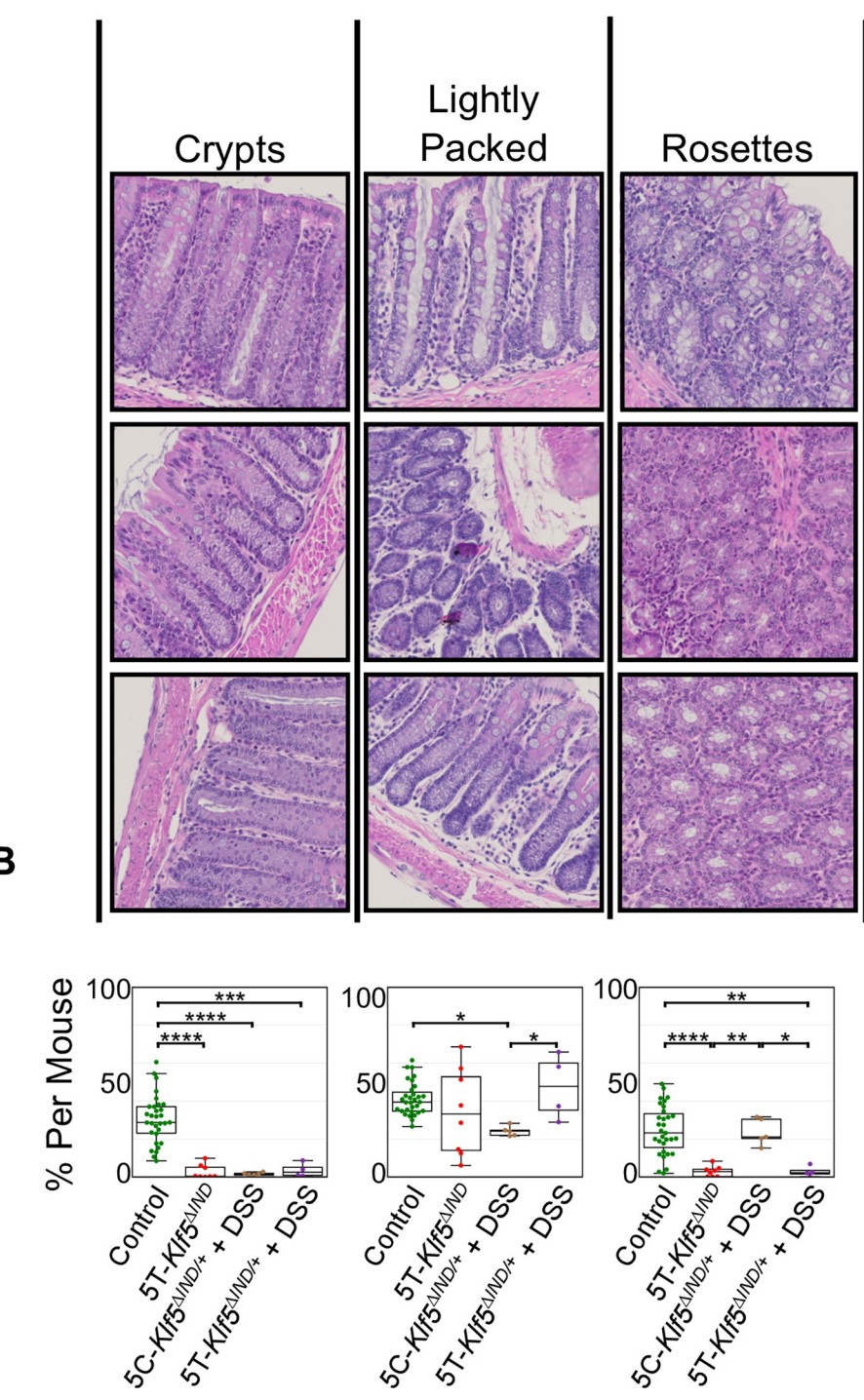

**Fig 3. Discovered 'Uninvolved' patch classes are enriched in control mice and show shifts in prevalence across colitis mouse models.** A) 'Uninvolved' patch classes with qualitative class labels. Patches most representative of class labels are shown. B) Box and whisker plots of 'Uninvolved' patch class proportions. Lines in center of box indicate median. Box boundaries refer to 1st and 3rd interquartile ranges (IQRs). Whiskers extend to furthest point within to 1.5*IQR. One–way ANOVA shows that these 'Uninvolved' patch classes are found in differing portions across colitis mouse models and controls. *$p < 0.0332$, **$p < 0.0021$, ***$p < 0.0002$, **** $p < 0.0001$.

not fully reflect histology. A future direction is thus to leverage this property of our pipeline to consider these borderline regions in the grading of swiss rolls.

The 'Mixed Pathology' k-means class identified during our second phase of model training (S2B–S2D Fig) motivated us to apply this approach to 'Involved' patch representations. K-means clustering identified four diseased patch classes (Fig 4A and 4B). By visual inspection and validation by a pathologist, these were classified as 'Inflammatory' (Milder, more heterogeneous phenotype with an influx of immune cell nuclei), 'Crypt Dropout' (loss of epithelium and displacement by stroma and immune cells), 'Crypt Dilation' (expansion of crypt lumen spaces), and 'Distorted Glands' (distortion of crypt structures). The 'Distorted Glands' persisted from the discovered k-means classes used in our second round of model training (S2C and S2D Fig). However, the 'Mixed Pathologies' class was replaced by new classes. Notably, immune cell infiltration, crypt dilation, and gland distortion have all been observed in human IBD [28–30]. Furthermore, these 'Involved' patch classes were present in significantly different proportions across our mouse models and controls (Fig 4B). Specifically, 5T-*Klf5*$^{\Delta IND}$ mice were enriched in the 'Inflammatory' and 'Crypt Dilation' classes, 5C-*Klf5*$^{\Delta IND/+}$ + DSS mice in the 'Inflammatory' and 'Crypt Dropout' classes, and 5T-*Klf5*$^{\Delta IND/+}$ + DSS mice in the 'Crypt Dropout' and 'Distorted Glands' classes (Fig 4B).

Stacked bar plots visually summarize these findings (Fig 4C). While all three 'Uninvolved' patch classes are present in control mice, 'Crypts' fall significantly for all three colitis mouse models and 'Lightly Packed' patches persist. 'Rosettes' are appreciable in 5C-*Klf5*$^{\Delta IND/+}$ + DSS and control mice. For 'Involved' classes, 'Inflammatory' and 'Crypt Dilation' patches are most prevalent in our 5T-*Klf5*$^{\Delta IND}$ mice. Given the shorter treatment course of this colitis mouse model (5 days), these may indicate more acute histological findings. On the other hand, the 'Crypt Dropout' class may be less acute as prevalence is higher in our longer treatment course 5C-*Klf5*$^{\Delta IND/+}$ + DSS (7 days) and 5T-*Klf5*$^{\Delta IND/+}$ + DSS mice (12 days). Similarly, 5T-*Klf5*$^{\Delta IND/+}$ + DSS mice are the only model with appreciable presence of 'Distorted Glands'. Distorted glands are one of the histological markers of chronic inflammation in human IBD [31]. Thus, this pipeline can histologically categorize colitis mouse models by which facets of human disease they recapitulate. In addition, a future direction is to partner this approach with immunohistochemical staining. This can shed further light on the relative contributions of length of colitis induction and immune cell presence in causing various types of histological abnormalities.

## Prediction of mouse model from H&E inputs

Next, we sought to address whether the variable presence of 'Uninvolved' patches and 'Involved' patch classes is sufficient to predict mouse model. We thus formed the pipeline detailed in S5A Fig to utilize 'Uninvolved' patch and 'Involved' k-means patch class proportions to train a linear determinant analysis (LDA) classifier to predict mouse model.

The LDA classifier was trained on the archived mouse cohort then applied to the prospective mouse cohort via the inference pipeline in S5B Fig. The classifier predicted which mouse model each swiss roll came from with an overall F1 score of 95.75% (Fig 4D). Of note, the k-means patch class stacked bar plots for the prospective cohort (S5C Fig) mirrored those for the archived mouse cohort (Fig 4C). As one of the 5T-*Klf5*$^{\Delta IND}$ mice was classified as control, we examined the input H&E. The WSI exhibited healthier appearing histology relative to WSIs from properly classified 5T-*Klf5*$^{\Delta IND}$ mice. This was apparent even from low magnification with decreased luminal space areas and more tightly packed crypts (S6 Fig). Although some 'Involved' areas were present in the 5T-*Klf5*$^{\Delta IND}$ mouse classified as control, crypts were more intact in these regions and accompanied with goblet cell presence. Goblet cells secrete mucus

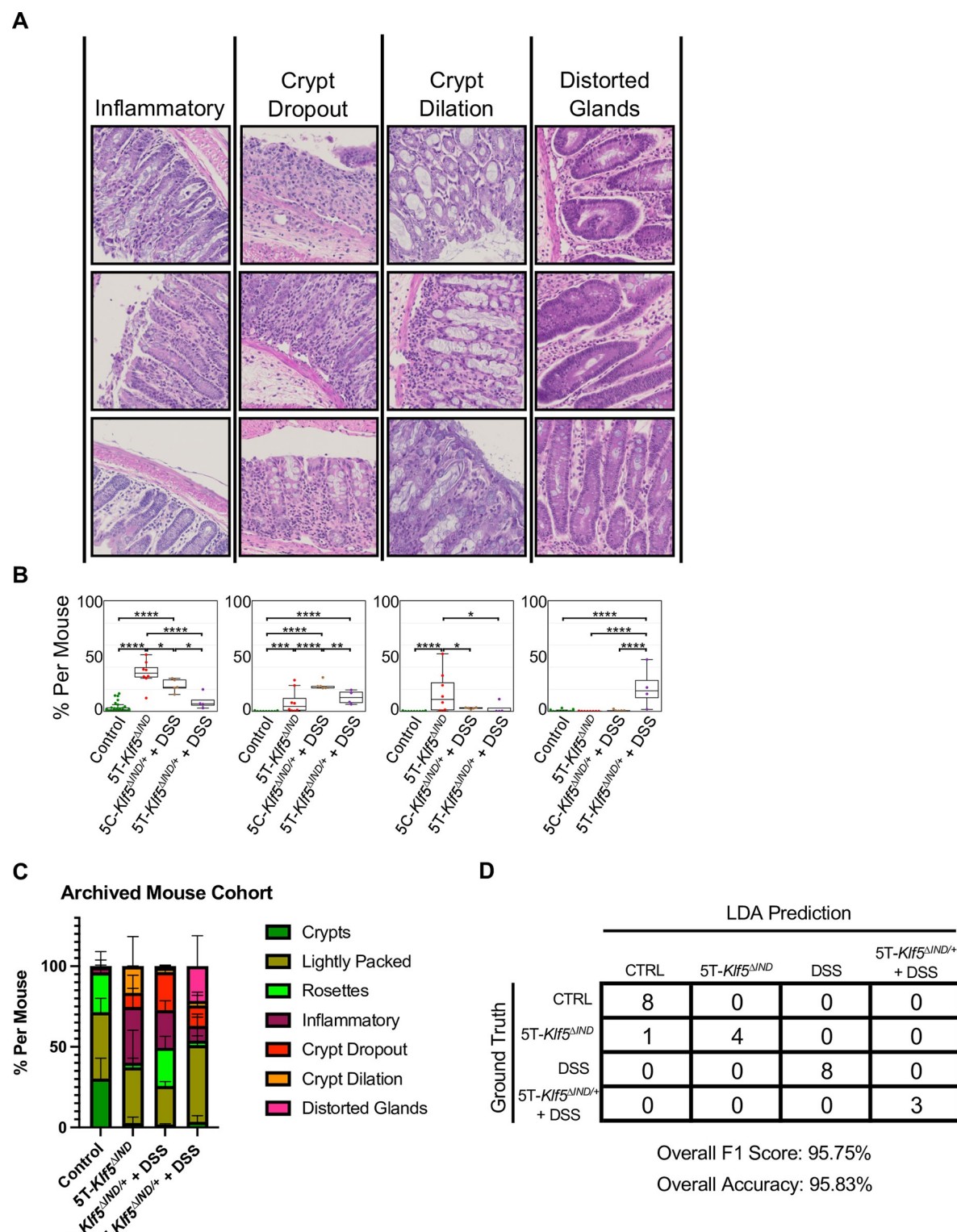

**Fig 4. Discovered 'Involved' k–means patch classes can be utilized by machine learning classifier to predict mouse model.** A) 'Involved' patch classes with qualitative class labels. Patches most representative of class labels are shown. B) Box and whisker plots of 'Involved' patch class proportions. Lines in center of box indicate median. Box boundaries refer to $1^{st}$ and $3^{rd}$ interquartile ranges (IQRs). Whiskers extend to furthest point within to $1.5^{*}$IQR. One–way ANOVA shows that these 'Involved' patch classes are found in differing portions across colitis mouse models and controls. $^{*}p<0.0332$, $^{**}p<0.0021$, $^{***}p<0.0002$, $^{****}$ $p<0.0001$. C) Stacked bar plot of total

'Uninvolved' and 'Involved' k–means patch class proportions across archived mouse cohort. Error bars show mean with standard deviation. D) Confusion matrix for linear determinant analysis (LDA) classifier mouse model predictions on prospective mouse cohort.

to provide a protective lining in the intestine, and decreased goblet cells are observed in CD and UC [32]. As such, this pipeline can histologically identify outlier mice with weaker colitis induction and phenotypes.

### 'Involved' patch presence predicts clinical score bins

We explored the correlation of 'Involved' predictions with clinical score. Mice were scored according to Cooper et al. [17]. Clinical scoring accounted for body weight loss, stool consistency, and blood in stool. For each prospective cohort mouse, we also calculated the InvolvedProportion, the proportion of overlay prediction pixels that were 'Involved'. InvolvedProportion shows moderate linear correlation with clinical score ($R^2$ = 0.669, S7A Fig).

We then assessed if 'Uninvolved' patch and "Involved" k-means patch class proportions provide value in predicting clinical score. Clinical scores were binned into "Low" (0–2), "Mid" (3–7) and "High" (8–12). The range for "Low" was set to 0–2, as control mice could still accumulate points for normal changes like <5% weight loss. The "Mid" and "High" bins each cover a range of 5 clinical score values. We trained a new LDA model on the archived mouse cohort to predict clinical score bins from patch proportions. This LDA model predicted clinical score bins in our prospective cohort with an accuracy of 87.5% and overall F1 score of 86.4% (S7B Fig). Two "Mid" mice were predicted as "High" and one "High" mouse was predicted as "Low". Of note, the two mice incorrectly predicted as "High" had the lowest proportion of 'Uninvolved' patches, while the mouse incorrectly predicted as "Low" had the highest. As such, there is overreliance on the proportion of 'Uninvolved' patches in predicting clinical score bin and a need to provide weights for the different histological classes. Although traditional scoring schemes like Cooper et al. [17] are based on similar concepts, they are often not developed across multiple colitis models and depend upon a simple tally of findings. Our computational approach can facilitate multi-mouse model comparisons through the quantification of histological findings across various colitis phenotypes. A future direction is thus to apply weakly-supervised approaches to determine weights for histological findings based on their relative contributions to phenotypic disease severity.

### Discussion

This study sought to address the simultaneous presence of colonic regions that are involved and uninvolved with abnormal histology within colitis mouse models. While these regions can be identified via careful inspection by a pathologist, doing so objectively over many samples is difficult and highly time-consuming. Deep learning models learn to associate class labels with implicit patterns in data and are well-suited for this task. A computational method to quantify the presence of abnormal histology can also serve as a histological readout for experiments performed on mice. One example would be to confirm the effect of treatments histologically as an accompaniment to clinical metrics like body weight, fecal blood, or stool consistency.

We thus present an 'Involved' versus 'Uninvolved' classifier for colons of 5T-*Klf5*$^{\Delta IND}$, DSS-treated, 5T-*Klf5*$^{\Delta IND/+}$ + DSS, and control mice. Specifically, we utilized a two-phase training approach. The initial phase uses mouse colitis status as patch ground truth labels, while the second phase incorporates feature extraction and clustering to generate improved, patch-level ground truth labeling. Importantly, our final model shows improved agreement with a pathologist relative to the model trained in just the initial phase of the approach.

In a related study, Bédard et al. published a proof of concept application of CNNs for the microscopic scoring of acute inflammation in DSS colitis [33]. They utilized a commercial artificial intelligence platform to detect muscle, normal mucosa, and acutely inflamed mucosa in H&E-stained murine colon WSIs. The authors calculated the ratio of acutely inflamed to total mucosa to assess the level of disease. Identifying these regions is an important step towards the microscopic grading of colitis samples. We similarly trained our Small Patch Classifier to detect 'Background', 'Muscle', 'Mucosa', and 'Submucosa' classes from 32x32 pixel patches. In our work, the Small Patch Classifier was implemented to improve our patch filtering process and to focus our 'Involved' versus 'Uninvolved' classifier overlays onto the 'Mucosa' and 'Submucosa' regions. A future direction is to explore whether the Small Patch Classifier outputs can improve our 'Involved' versus 'Uninvolved' model performance.

In an alternate approach, Rogers et al. attempted to segment colitis lesions in H&E-stained WSIs across the DSS, CD45RBHi, and IL-$10^{-/-}$ mouse models to grade disease severity [34]. The authors experienced segmentation challenges due to variabilities in crypt morphologies. They instead opted to generate a workflow based on CD3 immunohistochemical staining. Morphological variabilities likely increase more with disease relative to healthy histology. Here, we trained a patch-based 'Involved' versus 'Uninvolved' classifier. Including the task to classify the more homogenous 'Uninvolved' regions may help to separate out the 'Involved' regions. An additional benefit is that this has allowed us to perform clustering on just the 'Involved' patches to identify specific types of abnormal histology. Clustering on 'Uninvolved' patches also generated classes of normal histology. As with Rogers et al., we also incorporated multiple mouse models of colitis in our study. This has allowed for comparisons showing quantitative enrichment of histologic classes in different mouse models.

Specifically, we applied feature extraction, PCA-based dimensionality reduction and k-means clustering on 'Uninvolved' patches to define 'Crypt', 'Rosettes', and 'Lightly Packed' classes. Patches of the 'Crypt' class were significantly decreased in all colitis mouse models. The 'Rosettes' and 'Lightly Packed' classes were present in variable proportions across mouse models, mirroring shifts in proximal-involved and proximal-sparing patterns of injury. As the 'Lightly Packed' classes in colitis mice may represent regions of immune infiltration without overtly abnormal histology, we plan to assess the capacity of our model to consider these regions in the grading of colonic histology.

From 'Involved' patches, we identified four classes of abnormal histology–'Inflammatory', 'Crypt Dropout', 'Crypt Dilation', and 'Distorted Glands'. DSS-treated mice were enriched in the 'Inflammatory' and 'Crypt Dropout' classes, 5T-$Klf5^{\Delta IND}$ mice in the 'Inflammatory' and 'Crypt Dilation' classes, and 5T-$Klf5^{\Delta IND/+}$ + DSS in the 'Crypt Dropout' and 'Distorted Glands' classes. Swiss roll 'Uninvolved' patch and 'Involved' patch class proportions were sufficient to train an LDA classifier to predict mouse model and clinical score bins. Increased prevalence of the 'Inflammatory' and 'Crypt Dilation' classes in the shorter time course 5T-$Klf5^{\Delta IND}$ (5 days) and DSS-treated (7 days) mice may indicate these are more acute findings. Increases of 'Distorted Glands' in the combined induction model (12 days) and of 'Crypt Dropout' in the DSS-treated mice and combined induction model may reflect the more chronic nature of these histological classes. To the best of our knowledge, this is the first study to explore quantification of histological findings across mouse models to invite further assessment beyond qualitative descriptions. This may help validate the capacity of mouse models to capture certain aspects of human disease. To allow others to utilize the code, we have made available our prospective mouse cohort WSIs and the full inference pipeline from WSI scaling and patch extraction to LDA inference on github.

Our pipeline thus essentially examines swiss rolls, detects and categorizes histology, then predicts mouse model. As the 5T-$Klf5^{\Delta IND}$ mice have a Th17-mediated immune response [11], while the DSS model has been characterized as Th1-, Th17-, and innate immunity-mediated

[8, 26, 35], we have confirmed that the variable immune responses in these models are accompanied by differences in histology. The next step is to delineate the respective contributions of various immune populations to the presence of specific histological findings.

Accordingly, we believe this pipeline is the start for a spatially-motivated characterization of immune populations by histologic localization to 'Involved' and 'Uninvolved' regions. This approach may offer a complement to methods like flow cytometry and single-cell sequencing that can characterize immune responses with single-cell resolution but lose spatial context. A future direction is to map immune populations stained on serially sectioned slides to k-means patch classes to assess immune cell functionality and subtypes. Residence of immune populations in areas with and without disease may provide another angle towards characterization. Since many studies assessing functionality rely on knockouts or expensive neutralizing antibody experiments [11, 36–40], such a pipeline may offer a cheaper, quicker alternative.

One limitation of this method is that current application is restricted to the mouse models included in this study. However, we believe this is an important first step to establish and promote the potential of computer vision methodologies at the bench, and specifically, within the context of colitis mouse models. Another future direction is therefore to amplify the capacity of this method by incorporating other colitis models within training, such as the 2,4,6-Trinitrobenzenesulfonic acid (TNBS), IL-10 knockout, adoptive cell transfer, and oxazolone models [41]. Doing so would allow for the application of our method as a histological readout over a wider range of colitis mouse models.

More importantly, 'Involved' regions could then be extracted over an even more heterogeneous collection of colitis phenotypes. As these colitis models across literature differ in induction and flavor of immune responses, such an approach would open the door for synergy of knowledge gained from different mouse models. One possibility is that immune response characterizations, molecular approaches, and mechanistic studies focusing on pathophysiology could be linked to similar and dissimilar presence of types of histological findings across different murine colitis phenotypes. Notably, protein and genomic targets are often identified in preclinical animal models then evaluated for clinical value in patient specimens. Histological patterns identified in these animal models are driven by these same protein and genomic targets. Consequently, a future goal is to evaluate the clinical value and transferability of such histological phenotypes identified in animal models. This approach, therefore, has the potential to one day promote the discovery of novel therapeutic targets and pathways in IBD by serving as link between studies. As such, we believe that the integration of computational and computer vision approaches offers significant and exciting potential in bringing together the depth of human knowledge that has been gained from across colitis mouse models and in encouraging further collaboration across groups.

## Materials and methods

### Mice

All studies involving mice were approved by the Stony Brook University (SBU) Institutional Animal Care and Use Committee (IACUC). All mice were house in the SBU Division of Laboratory Animal Resources (DLAR) and maintained on a 12:12 hour light-dark cycle. The DLAR facility has optimized conditions regarding well-regulated temperature and humidity and light settings to ensure a stable, reproducible environment for animal growth. All mice carried an inducible Cre recombinase gene under the *Villin* promoter. These mice carried either two wild-type alleles of *Klf5* (*Villin-CreER^{T2};Klf5^{+/+}*) or were heterozygous or homozygous for an additional *Klf5* allele flanked by loxP sites (heterozygous: *Villin-CreER^{T2};Klf5^{ΔIND/+}*, homozygous: *Villin-CreER^{T2};Klf5^{ΔIND/ΔIND}*).

Treatment schedules are detailed in S1A Fig. All treatment conditions are available in S1 Table. Littermates were split across control groups in experiments to minimize littermate-specific batch effects. S.K. was to only one to know of treatment group allocation. For all experiments, mice were eight to ten weeks old and female, as we have seen increased Cre recombination efficiency relative to males in the 5T-$Klf5^{ΔIND}$ model [11]. Intraperitoneal (IP) injection of tamoxifen at 1mg/day dissolved in corn oil for 5 days was performed as previously described [11]. Control mice received 5 days of IP corn oil injections. For dextran sodium sulfate (DSS) treatment, mice in the archived cohort received 2.5% DSS in drinking water. In the prospective cohort, DSS-treated mice received 3% DSS, as we observed this was the optimal concentration for experiments at the Stony Brook University facilities. Controls for DSS treatments were provided normal drinking water. For combined induction, $Klf5^{ΔIND/+}$ mice received 2.5% DSS following 5 days of IP injection of tamoxifen at 1mg/day dissolved in corn oil. Upon conclusion of treatments, whole colons were collected and swiss-rolled according to [16]. Swiss-rolled colons were then formalin-fixed and paraffin-embedded (FFPE). Mice receiving histological or clinical scoring were scored according to Cooper et al. [17]. Histological scoring accounted for inflammatory cells in the lamina propria, crypt damage, and extent of ulceration. Clinical scoring covered weight loss, stool consistency, and fecal blood.

To minimize distress, daily observations of body weight and abnormalities, including anorexia, rectal prolapse, intractable diarrhea, ruffled fur, labored breathing, hunched posture, or lack of normal investigative behavior were performed. Any mouse exhibiting these observations or experiencing greater than 15% loss of baseline weight during treatment course were euthanized. Euthanasia was performed by delivering carbon dioxide via compressed gas followed by immediate cervical dislocation.

## Whole slide image (WSI) generation

FFPE swiss-rolled colons were sectioned onto glass slides and stained by hematoxylin and eosin (H&E) [16]. These glass slides were then scanned and digitized at 40X magnification (0.17 µM/pixel) to a.vsi format by the Olympus VS120 Digital Virtual Slide System (VS120-L100-W). The files were converted to a tiff format for further downstream use.

## H&E 'Involved' versus 'Uninvolved' classifier training

We trained the ResNet-34 (RN-34) neural network, which is considered a landmark architecture for its introduction of skip connections [21]. Briefly, skip connections allow the model to 'skip' parts of its architecture where training would typically be impeded due to gradient explosion or vanishing [22]. We trained a RN-34 model pretrained on ImageNet, a large dataset with natural images comprising 1000 classes [23].

An overview schematic of our two-phase training approach is available in S2A and S2B Fig. We downsize WSIs by a factor of 8 and extract 224x224 pixel patches. Patches containing too little tissue are filtered out. The initial phase (S2A Fig) utilizes mouse colitis induction status as ground truth to label extracted patches. As such, colonic regional heterogeneity (Fig 1A) is not addressed here. Instead, all patches from colitis mice are provided a 'Colitis' label, while all patches from control mice are provided a 'Control' label. Once labeled, 70% of patches were used to train the model, 10% of patches were used to assess performance mid-training, and 20% of patches were used as a held-out test set.

To generate more granular, patch-level ground truth labeling, we pursued the approach in S2B Fig. We used the initial phase model as a feature extractor to convert all archived mouse cohort image patches to high-dimensional 512-length numerical representations learned

during the initial phase of training. K-means clustering on these representations revealed patch classes (S2C Fig).

We then utilized these k-means clusters to generate new patch-level ground truth labels. As our control mice did not receive any colitis induction, we provided 'Uninvolved' ground truth labels to all patches from these mice. For our mice with colitis induction, we referred to the k-means clustering results. With this new ground truth labeling, we trained another RN-34 model pretrained on ImageNet [23] using a 70/10/20% training/validation/test set split.

### Small patch classifier training for preprocessing

From our archived mouse cohort WSIs, we extracted 32x32 pixel patches and trained an additional RN-34 classifier pretrained on ImageNet [23] (S3 Fig). Our Small Patch Classifier predicts these patches as one of the 'Background', 'Muscle', 'Tissue', and 'Submucosa' classes.

### Tissue map generation and patch filtering

For every H&E-stained, swiss-rolled colon WSI, we first extract 32x32 pixel patches. The Small Patch Classifier is applied to each to generate a tissue map for every input H&E WSI (S4A Fig). The tissue map is of the same dimension as the input H&E. White areas represent 'Tissue' and 'Submucosa' classifications, while the black areas represent 'Muscle' and 'Background'. For each subsequently extracted 224x224 pixel H&E patch, the corresponding area is extracted from the tissue map. Each patch is then thresholded to evaluate whether enough tissue or submucosa is present. Practically, we implement a 65% threshold. Patches with more than 65% of 'Background' or 'Muscle' regions are filtered out, while those with less are kept (S4B Fig).

### Overlap patch extraction and overlay generation

To reduce classification dependence for what patch happened to be extracted from a region in a WSI, we extracted overlapping patches. Beginning with existing patch locations, we iterated 20 pixels 10 times in each direction (up/down/left/right) and extracted new patches. We then again filtered out patches containing too little tissue. This led to an increase of ~450 to ~200,000 patches per mouse.

To generate overlays, our 'Involved' versus 'Uninvolved' classifier was applied to all patches, including overlapping patches. At every pixel, 'Involved' and 'Uninvolved' prediction confidences were averaged. Red areas correspond to pixels with >50% confidence of 'Involved' predictions, while green areas correspond to pixels with >50% confidence of 'Uninvolved' predictions. Furthermore, all regions in tissue maps corresponding to 'Muscle' and 'Background' were eliminated from the overlay. This generated final outputs with predictions over only mucosal and submucosal areas (Fig 2B and 2C).

### Discovery of patch classes

Overview schematic is shown in S5A Fig. From our archived mouse cohort, we extracted all patches, including overlapping patches, classified as 'Involved'. We utilized our final 'Involved' versus 'Uninvolved' classifier as a feature extractor, then performed PCA-based dimensionality reduction to 250 principal components (PCs) that capture 95% of variability. We then applied k-means clustering to identify disease patch classes. This process was repeated for all patches classified as 'Uninvolved' from our archived mouse cohort to identify healthy patch classes. PCA-based dimensionality reduction for 'Uninvolved' patches required 255 PCs to account for 95% of variability.

## Machine learning classifier to predict mouse model

Per mouse proportions of 'Uninvolved' patches and the 'Involved' k-means patch classes were generated for every sample from the archived mouse cohort. These proportions were utilized to train a linear determinant analysis (LDA) classifier to predict mouse model (S5A Fig). For inference, the schematic is shown in S5B Fig. Overlapping patches are extracted from our prospective mouse cohort samples and filtered according to tissue maps generated by the small patch classifier. Our 'Involved' versus 'Uninvolved' classifier is applied, and 'Involved' patches undergo a further round of PCA-based dimensionality reduction that is first fit to archived mouse cohort data. Our k-means model, which is also trained on our archived mouse cohort, then categorizes 'Involved' patches into one of the discovered classes. Finally, the per mouse proportions of 'Uninvolved' patches and 'Involved' k-means patch classes are generated for each mouse and fed into the LDA classifier to infer the mouse model.

## Machine learning classifier to predict clinical score bins

The generated per mouse proportions of 'Uninvolved' patches and the 'Involved' k-means patch classes were also utilized to train a separate LDA classifier to predict clinical score bins. Clinical score bins were "Low" (0–2), "Mid" (3–7), and "High" (8–12). The "Low" range was selected to account for normal observations that might accumulate clinical score, such as <5% weight loss over experimental course. This LDA classifier was trained on the archived mouse cohort and applied to predict clinical score bins in the prospective mouse cohort.

## Statistics

One-way ANOVA and student's t-test calculations were performed on GraphPad Prism Version 9.0.0 for Mac. Linear correlation $R^2$ values were generated in Python. One-way ANOVA was performed when there were three or more comparison groups. Student's t-test was performed for two comparison groups.

## Supporting information

**S1 Table. Mouse cohort sample numbers per genotype and treatment.**
(TIF)

**S1 Fig. Colitis mouse models.** A) Treatment schedules for each mouse model. DSS model in prospective cohort has no injections to confirm abnormal pathology is recognized independent of corn oil. Additionally, 3.0% DSS was used in prospective mice, as this was observed to be the optimal concentration at the Stony Brook facilities. B) Swiss roll of $Klf5^{WT}$ ($Villin$-$CreER^{T2};Klf5^{+/+}$) mouse treated with DSS and no TAM or CO injections. C) Though not used to train our classifier, swiss rolls of corresponding controls (5T-$Klf5^{ΔIND/+}$ and 5C-$Klf5^{ΔIND/+}$) are shown for combined colitis model. D) Clinical scores combining weight loss, stool consistency, and fecal blood according to Cooper et al. [17] for combined colitis model, 5C-$Klf5^{ΔIND/+}$ + DSS, and control mice. E) Histological scores according to Cooper et al. [17]. One-way ANOVA was performed for D) and E). $^*p<0.0332$, $^{**}p<0.0021$, $^{***}p<0.0002$, $^{****}$ $p<0.0001$. (TIF)

**S2 Fig. Second phase of training with k-means patch class labels improves model performance.** A) Overview schematic showing initial phase of RN-34 model training that uses mouse colitis status as patch ground truth labels. Thus, intracolonic heterogeneity in colitis mice is not addressed at this round of patch labeling. B) Second phase of model training that uses trained RN-34 model from A) as a feature extractor for patches in dataset. K-means

clustering on extracted features generated patch classes. Student t-test is utilized to assess whether patch classes are significantly enriched in colitis or control mice. Patches from control mice are labeled as 'Uninvolved', as no colitis induction occurred. For colitis mice, patches are labelled 'Uninvolved' or 'Involved' according to k-means predictions. C) K-means patch classes identified during second phase of training in B) used for ground truth labeling. D) Box and whisker plots of patch class proportions. Lines in center of box indicate median. Box boundaries refer to $1^{st}$ and $3^{rd}$ interquartile ranges (IQRs). Whiskers extend to furthest point within to 1.5*IQR. Student's t-tests were performed. *$p<0.05$, **$p<0.01$, ***$p<0.001$, ****$p<0.0001$. E) Rosettes are in DSS-treated mice have higher means relative to other colitis models. One-way ANOVA shows a statistically significant difference between groups. F) Independent test set output confusion matrix for initial phase model. G) 200 patches from 4 mice (1 control, 1 of each colitis mouse model) were labeled as 'Uninvolved' or 'Involved' by a pathologist. 29/200 patches were discarded for not enough spatial context to provide a label. Inference using models trained in A) and B) show that the k-means patch labeling approach increased prediction agreement with pathologist-generated labels.
(TIF)

**S3 Fig. Small patch classifier for preprocessing.** A) Example 32x32 pixel patches for the 'Background', 'Tissue', 'Muscle', and 'Submucosa' classes used to train the Small Patch RN-34 Classifier. B) Trained Small Patch Clasifier confusion matrix outputs for independent test set of 4 mice, each with 100 patches of each class (1600 total patches). C) Example overlay of Small Patch Classifier on DSS-treated test set mouse.
(TIF)

**S4 Fig. Tissue map generation and filtering process.** A) Small Patch Classifier from S3 Fig is applied to all 32x32 pixel patches extracted from a WSI. A tissue map is generated where 'Background' and 'Muscle' are black, while 'Tissue' and 'Submucosa' are white. The yellow arrow indicates a portion of muscle that is assigned to the Background/Muscle class on the corresponding tissue map. B) Example 224x224 pixel H&E patches with corresponding tissue map patches and filtering decisions. For each extracted patch, the decision is made based on whether there is more than 65% (filter) or less than 65% (keep) of unwanted Background/Muscle area on the corresponding tissue map patch.
(TIF)

**S5 Fig. Linear determinant analysis classifier training and inference schematics for mouse model predictions.** A) Overview schematic. All patches classified as 'Involved' by our model, including overlapping patches, undergo feature extraction by our final 'Involved' versus 'Uninvolved' classifier. Subsequent PCA-based dimensionality reduction and k-means clustering identify 4 'Involved' patch classes (Fig 4A). An LDA classifier is trained on per mice proportions of 'Uninvolved' patches and 'Involved' k-means patch classes to predict mouse models. B) Overview schematic for inference pipeline. Overlapping patches are extracted and patches containing too much background or muscle are filtered out via the tissue map process in S4 Fig. The classifier is applied to each kept patch. 'Involved' patches undergo further RN-34 feature extraction and PCA-based dimensionality reduction. These patches are then classified 'into one of the four 'Involved' patch classes. The trained LDA model then predicts mouse model from the per-mouse proportions of 'Uninvolved' patch and 'Involved' k-means patch classes. C) Stacked bar plot of total 'Uninvolved' and 'Involved' k-means patch class proportions across prospective mouse cohort. Error bars show mean with standard deviation.
(TIF)

**S6 Fig. 5T-*Klf5$^{\Delta IND}$* swiss roll predicted as control has healthier appearing histology than 5T-*Klf5$^{\Delta IND}$* swiss roll predicted as 5T-*Klf5$^{\Delta IND}$*.** Compared to the properly predicted 5T-*Klf5$^{\Delta IND}$* swiss rolled colon (left), the 5T-*Klf5$^{\Delta IND}$* colon predicted as control has fewer histological abnormalities and represents a mouse with weak colitis induction. Yellow arrows indicate healthy goblet cells. Red arrows indicate absence of goblet cells. Orange arrows indicate crypt loss.
(TIF)

**S7 Fig. 'Involved' predictions likely provide value in predicting clinical score.** A) Scatter plot of Clinical Score verse InvolvedProportion. Clinical scores were obtained according to Cooper et al. [17]. InvolvedProportion is the proportion of 'Involved'-predicted pixels out of all prediction pixels in overlays. B) LDA trained on archived mouse cohort predicts prospective mouse cohort clinical score bins from per-mouse 'Uninvolved' patch and 'Involved' k-means patch class proportions. Clinical score bins are "Low" (0–2), "Mid" (3–7), and "High" (8–12).
(TIF)

**S1 File. Full arrive 2.0 guidelines checklist.**
(PDF)

## Acknowledgments

The authors wish to acknowledge the Stony Brook Research Histology core for their expert assistance with paraffin embedding, sectioning, and performing H&E staining on our colon samples. The authors also wish to acknowledge the Stony Brook Division of Laboratory Animal Resources for their expert assistance with caring, housing, and maintaining mice used to generate results included in this manuscript.

## Author Contributions

**Conceptualization:** Soma Kobayashi, Agnieszka B. Bialkowska, Joel H. Saltz, Vincent W. Yang.

**Data curation:** Soma Kobayashi, Jason Shieh, Ainara Ruiz de Sabando, Julie Kim, Yang Liu, Sui Y. Zee, Prateek Prasanna, Agnieszka B. Bialkowska, Joel H. Saltz, Vincent W. Yang.

**Formal analysis:** Soma Kobayashi, Jason Shieh, Ainara Ruiz de Sabando, Julie Kim, Yang Liu, Sui Y. Zee, Prateek Prasanna, Agnieszka B. Bialkowska, Joel H. Saltz, Vincent W. Yang.

**Funding acquisition:** Joel H. Saltz, Vincent W. Yang.

**Investigation:** Soma Kobayashi, Jason Shieh, Ainara Ruiz de Sabando, Julie Kim, Yang Liu, Sui Y. Zee, Prateek Prasanna, Agnieszka B. Bialkowska, Joel H. Saltz, Vincent W. Yang.

**Methodology:** Soma Kobayashi, Ainara Ruiz de Sabando, Julie Kim, Yang Liu, Sui Y. Zee, Prateek Prasanna, Agnieszka B. Bialkowska, Joel H. Saltz, Vincent W. Yang.

**Project administration:** Soma Kobayashi, Agnieszka B. Bialkowska, Joel H. Saltz, Vincent W. Yang.

**Resources:** Sui Y. Zee, Prateek Prasanna, Agnieszka B. Bialkowska, Joel H. Saltz, Vincent W. Yang.

**Software:** Soma Kobayashi, Prateek Prasanna, Joel H. Saltz, Vincent W. Yang.

**Supervision:** Prateek Prasanna, Agnieszka B. Bialkowska, Joel H. Saltz, Vincent W. Yang.

**Validation:** Soma Kobayashi, Sui Y. Zee, Prateek Prasanna, Agnieszka B. Bialkowska, Joel H. Saltz, Vincent W. Yang.

**Visualization:** Soma Kobayashi, Sui Y. Zee, Prateek Prasanna, Agnieszka B. Bialkowska, Joel H. Saltz, Vincent W. Yang.

**Writing – original draft:** Soma Kobayashi, Agnieszka B. Bialkowska, Joel H. Saltz, Vincent W. Yang.

**Writing – review & editing:** Jason Shieh, Ainara Ruiz de Sabando, Julie Kim, Yang Liu, Sui Y. Zee, Prateek Prasanna, Agnieszka B. Bialkowska, Joel H. Saltz, Vincent W. Yang.

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
