## [Decision Letter · Decision Letter 0]

13 Jul 2022

PONE-D-22-13584Deep learning-based approach for the characterization and quantification of histopathology in mouse models of colitisPLOS ONE

Dear Dr. Yang,

Thank you for submitting your manuscript to PLOS ONE. After careful consideration, we feel that it has merit but does not fully meet PLOS ONE’s publication criteria as it currently stands. Therefore, we invite you to submit a revised version of the manuscript that addresses the points raised during the review process.

This is a comprehensive study by the authors. Please provide point-by-point response to each of the reviewers comments. 

We look forward to receiving your revised manuscript.

Kind regards,

Sripathi M Sureban, Ph.D.

Academic Editor

PLOS ONE

Journal Requirements:

2. As part of your revision, please complete and submit a copy of the Full ARRIVE 2.0 Guidelines checklist, a document that aims to improve experimental reporting and reproducibility of animal studies for purposes of post-publication data analysis and reproducibility: https://arriveguidelines.org/sites/arrive/files/Author%20Checklist%20-%20Full.pdf (PDF). Please include your completed checklist as a Supporting Information file. Note that if your paper is accepted for publication, this checklist will be published as part of your article.

“This work is support by grants from the National Institutes of Health: DK052230 to V.W.Y. and CA205109/CA225021 to J.H.S.”

“NIH grants:

DK052230 to V.W.Y. and CA205109/CA225021 to J.H.S.

Additional Editor Comments:

This is a comprehensive study by the authors and please address that concerns by the each of the reviewers. Also, provide point-by-point response for easy review.

Reviewers' comments:

Reviewer's Responses to Questions

**Comments to the Author**

1. Is the manuscript technically sound, and do the data support the conclusions?

Reviewer #1: Yes

Reviewer #2: Yes

2. Has the statistical analysis been performed appropriately and rigorously? 

Reviewer #1: Yes

Reviewer #2: Yes

3. Have the authors made all data underlying the findings in their manuscript fully available?

Reviewer #1: Yes

Reviewer #2: Yes

4. Is the manuscript presented in an intelligible fashion and written in standard English?

Reviewer #1: Yes

Reviewer #2: Yes

5. Review Comments to the Author

Reviewer #1: Mouse modeling is an important research tool in the study of colitis pathophysiology. Colitis induced by putting the chemical DSS in drinking water is a widely adapted model that results in disruption of the epithelium particularly in the distal colon. Other models may lead to more proximal colitis (as in the case of mice with Klf5 KO in IECs as presented in this paper. Thus, different models exhibit different features of colitis and at different locations within the colon. This variability often leaves research groups with less experience in this models to inappropriately score colitis by picking the wrong or a very limited region that may not reflect the overall colitis severity. In this way, having an automated method of scoring murine colitis models may improve overall accuracy and validity of studies as well as hasten the evaluation time required for many mice.

In this study by Kobayashi, Yang, et all, they apply an deep learning approach to quantify sections of the H&E-stained colon slides with or without colitis in the acute DSS, Klf5-KO, and combo models. Overall, the authors go into appropriate depth of describing their methodology and results with some exceptions described below. The authors also acknowledge previous efforts at developing similar deep-learning algorithms and presented an improved algorithm using two-step training approach to decipher involved and uninvolved regions of colitis in the mouse gut. As the authors admit, the usefulness of this approach will be improved once it is extended to more commonly used modles (T-cell transfer, TNBS, IL-10KO, etc) as the selection of the less used Klf5 is based on the authors own familiarity with this less widely used model. Still, this paper provides a beginning framework for what may prove to be a useful tool in standardizing the assessment of histologic disease severity in mouse modeling of colitis. Whether the tool will eventually be able to reveal histologic/pathologic links and thus identify new therapeutic targets in colitis remains to be determined.

Specific comments to be addressed/clarified:

1. I found the manuscript to be overly verbose in several sections. Consider consolidating some of the background and discussion as well as repetitive comments in the results.

2. Sup Fig 1: The histology score and clinical score of the 5T-Klf5∆IND/+ is not significantly different from controls. The authors should address the impact of this on the overall model performance.

3. Also, there is no evidence from figure S1D to back up this statement: “Clinical scoring metrics combining weight loss, stool consistency, and fecal blood (16) show significant increases for 5T-Klf5∆IND/+ + DSS mice relative to 5C- Klf5∆IND/+ + DSS”. The authors should clarify.

4. The authors should provide an explanation for why the combined induction model (5T-Klf5∆IND/+ + DSS mice) has lower histological score (Fig S1D) and extent of ulceration compared to either of the models with one induction only.

5. The authors should explain why different statistical tests – Student T test vs One-way ANOVA were used in fig S1D vs S1E fig. It is also unclear if multiple comparison was accounted for. A statistics section in Methods will make this clearer.

6. Please add a graphical model of the pattern of injury in the combined induction model in Figure 1B.

7. Consider assessing the performance of the model on theDSS chronic colitis model and assess if it will be characterized by “involved patches” histology similar to the patterns seen with 5T-Klf5∆IND/+ + DSS model. (Possible?)

8. Clarify if “Rosette” class patch used as “ground truth label” in the second phase of training? If yes, did the authors should consider excluding this class given that it was not a good discriminator of normal vs abnormal from the first training phase.

9. The authors should clarify the basis for using 65% as the threshold to filter ‘Background’ or ‘Muscle’ regions as shown in fig S4B.

10. The representative image of 5T-Klf5∆IND shown in Figure 2B does not seem to support the statements by the authors, and the graphical depiction in Fig 1B, as it appears the most involved area in the representation image in Fig 2B is the distal colon.

11. If feasible, it will be interesting to see the performance of a model generated using just 2 of the three models (e.g 5T-Klf5∆IND/+ and DSS model., and tested in the third model (5T-Klf5∆IND/+ + DSS model). This would serve as development and validation approach.

12. Please report if there is a correlation between higher proportion of “involved patches” identified using the final model and the clinical score in each individual mouse?

13. From figure 4c, of all the three models, 5T-Klf5∆IND/+ appear to have the highest proportion of patches representing “involved” area. This does not correlate with the lower clinical and histologic scores shown in SFig 1D and 1E. Please explain.

Reviewer #2: This manuscript identifies characterization and quantification of histopathology in mouse models of colitis. The authors performed three different mouse model of colitis such as dextran sodium sulfate (DSS) model, a genetic model of Klf5ΔIND (Villin-CreERT2;Klf5fl/fl36 ), and a combination of both induction methods.

The analysis of colitis in all three models using Convolutional neural networks (CNNs). CNNs was deployed to identify Involved’ & ‘Uninvolved’ region of colon for characterization of colitis which is unique. A computational method technique gives an added advantage to pathologists for accurate interpretation and identification of colitis.

The results indicate that trained classifier permits for mining of involved and uninvolved colonic regions across mice cluster to determinant and distinguish colitis more accurately.

In overall manuscript is acceptable for publication, presenting reasonable data supporting the conclusion, appropriate statistical analysis.

Minor comments.

How did the authors collect the colon tissue? Was it from proximal colon and distal colon for the histo-analysis?

Beside histology did the authors used any markers to confirm colitis?

Some parts of the introduction were not clear perhaps and should be rewritten; one such example “Introduction, line 61 is not clear “Groups have utilized methods such as flow cytometry and single-cell RNA sequencing to characterize the immune responses in these mice” what do the authors mean?

6. PLOS authors have the option to publish the peer review history of their article (what does this mean?). If published, this will include your full peer review and any attached files.

Reviewer #1: No

Reviewer #2: No

---

## [Decision Letter · Decision Letter 1]

12 Aug 2022

Deep learning-based approach to the characterization and quantification of histopathology in mouse models of colitis

PONE-D-22-13584R1

Dear Dr. Yang,

We’re pleased to inform you that your manuscript has been judged scientifically suitable for publication and will be formally accepted for publication once it meets all outstanding technical requirements.

Kind regards,

Sripathi M Sureban, Ph.D.

Academic Editor

PLOS ONE

Additional Editor Comments (optional):

The review panel congratulates the authors on their extensive work and novel methodology to interpret the colitis model.

Reviewers' comments:

Reviewer's Responses to Questions

**Comments to the Author**

1. If the authors have adequately addressed your comments raised in a previous round of review and you feel that this manuscript is now acceptable for publication, you may indicate that here to bypass the “Comments to the Author” section, enter your conflict of interest statement in the “Confidential to Editor” section, and submit your "Accept" recommendation.

Reviewer #1: All comments have been addressed

Reviewer #2: All comments have been addressed

2. Is the manuscript technically sound, and do the data support the conclusions?

Reviewer #1: Yes

Reviewer #2: Yes

3. Has the statistical analysis been performed appropriately and rigorously? 

Reviewer #1: Yes

Reviewer #2: Yes

4. Have the authors made all data underlying the findings in their manuscript fully available?

Reviewer #1: Yes

Reviewer #2: Yes

5. Is the manuscript presented in an intelligible fashion and written in standard English?

Reviewer #1: Yes

Reviewer #2: Yes

6. Review Comments to the Author

Reviewer #1: Thank you Drs. Kobayashi, Yang and team for your excellent revision and thorough responses. thank you also for your work advancing this approach which hopefully will help in moving the field forward through greater standardization in interpretation of colitis modeling.

Reviewer #2: In the revised version of the manuscript the authors addressed all the queries raised by the reviewer. I believe the manuscript should be considered for the publication in PLOS ONE.

7. PLOS authors have the option to publish the peer review history of their article (what does this mean?). If published, this will include your full peer review and any attached files.

Reviewer #1: **Yes: **Matthew Ciorba

Reviewer #2: No

---

## [Editor Report · Acceptance letter]

18 Aug 2022

PONE-D-22-13584R1 

Deep learning-based approach to the characterization and quantification of histopathology in mouse models of colitis. 

Dear Dr. Yang:

I'm pleased to inform you that your manuscript has been deemed suitable for publication in PLOS ONE. Congratulations! Your manuscript is now with our production department. 

Kind regards, 

on behalf of

Dr. Sripathi M Sureban 

Academic Editor

PLOS ONE